# A Scoping Review of Nutrition Health for Older Adults: Does Technology Help?

**DOI:** 10.3390/nu15204402

**Published:** 2023-10-17

**Authors:** Dara L. LoBuono, Michael Milovich

**Affiliations:** 1Department of Health and Exercise Science, School of Nursing and Health Professions, Rowan University, James Hall Room 1035, 201 Mullica Hill Road, Glassboro, NJ 08028, USA; 2Department of Marketing and Business Information Systems, Rohrer College of Business, Rowan University, Business Hall Room 316, 201 Mullica Hill Road, Glassboro, NJ 08028, USA; milovich@rowan.edu

**Keywords:** digital health, geriatrics, nutrition, older adults, technology, literature review

## Abstract

The technological developments in healthcare may help facilitate older adult nutritional care. This scoping review includes research in technology and nutrition to (1) explain how technology is used to manage nutrition needs and (2) describe the forms of technology used to manage nutrition. Five major databases were the foundation for papers published from January 2000 to December 2020. The most common type of technology used is software to (1) “track, plan, and execute” nutrition management and for (2) “assessing” technology use. “Track, plan, and execute” includes tracking food intake, planning for changes, and executing a plan. “Assessing” technology use is collecting nutrition data from a provider’s or an older adult’s self-use of technology to understand dietary intake. Hardware is the second most type of technology used, with tablet computers for software and internet access. The findings reveal that software for older adults lacks standardization, the Internet of Things is a promising area, the current device emphasis is the tablet computer, and broadband internet access is essential for nutrition care. Only 38 studies were published in the last five years, indicating that nutrition management for older adults with hardware or software has not reached a significant research mass.

## 1. Introduction

The percentage of older adults aged 65 and older increases daily [1]; however, not all older adults are able to access technology. Ten percent of the United States (U.S.) population does not use the internet; 25 percent are 65 years and older [2]. Yet, studies show that computer-based touch-screen assessment systems [3] and robots with specific responsiveness to diet [4] improve nutrition awareness among older adults. Older adults navigate technology differently compared with other age groups [5], and consideration for this difference should be considered in helping this population adopt new technology to improve their quality of life (QOL). Including older adults in the mainstream inclusion of technology and new designs could help decrease the risk of age-related health conditions, such as weight loss from inadequate nutritional intake [6].

### 1.1. Biological Aging and Nutrition

Physiological changes in the cardiovascular, neurological, respiratory, and musculoskeletal systems are a natural part of biological aging and are a primary factor of age-related chronic illness [7]. These age-related conditions increase an older adult’s risk for poor nutritional status [8]. Age-related changes in appetite, ability to taste and smell, and level of food involvement [9] can further worsen nutrition status, chronic diseases, disabilities, and QOL [10]. Optimal dietary intake can increase lifespan [7,11,12,13,14] and independence and minimize healthcare costs [15,16]. Therefore, health status can vary widely among older adults, and technology may help better manage variability among older adults as they age and optimize diet quality [17]. A registered dietitian nutritionist (RDN), an expert in food and nutrition, can provide evidence-based nutrition recommendations and guidance to address barriers older adults may face in achieving optimal dietary intake.

### 1.2. The Nutrition Care Process

The nutrition care process (NCP) is a systematic framework that RDNs use to provide individualized, high-quality medical nutrition therapy and is composed of four interconnected steps: (1) assessment (document nutrition and medical history; biochemical data; medical tests; anthropometrics; and nutrition-focused physical examination); (2) diagnosis (use the collected data to make a diagnosis), (3) intervention (select targeted interventions and goals); and (4) monitoring/evaluation (ensure goal completion [8]). An RDN can work with an older adult to detect and treat nutrient deficiencies, improve or maintain body composition, manage diet-related health conditions, and enroll in food assistance programs [18]. Technology can enhance the management of this framework.

### 1.3. Accessing Nutrition Professionals and Services

The exponential increase in the aging population has encouraged “aging in place”, a practice where older adults live in their community homes independently, safely, and comfortably [19]. Up to 98% of older adults are community-dwelling and live outside nursing homes [20]. Healthy dietary patterns can facilitate “aging in place” by helping to prevent and/or manage chronic diseases while also helping to optimize physical and cognitive functioning [15,16,21]. Efforts to address nutritional status among older adults include government-funded meal support and health support programs [22,23]. Additionally, RDNs can help older adults optimize their nutrition status. However, older adults face barriers in accessing an RDN and nutrition services, which include residing in rural areas, disabilities and mobility issues, low socioeconomic status, and accessibility to transportation or health care services [24,25]. Some older adults lack knowledge of unintended weight loss’s negative consequences on their health [26,27]. Others face psychosocial barriers, like loneliness and isolation, which can affect their desire to eat, and some older adults are unwilling to be screened for malnutrition [28]. In the U.S., federal medical insurance for older adults covers nutrition therapy and group education classes only for specific diseases [29,30]. As a result, one-on-one nutrition counseling sessions and education programs are limited to subsets of older adults [31,32]. Technology-based interventions are promising strategies to overcome barriers older adults face and help them access nutrition services [23]. The technology could bring value to conducting nutrition assessments, promoting healthy eating, and improving dietary quality and nutritional status [33,34,35]. As a result, incorporating technology in the care of older adults may enhance nutrition care and status, promote “aging in place”, and improve QOL [6,23].

### 1.4. Technology for Aging

Caregivers have introduced technologies to assess, track, and manage older adults’ health [36,37] along with promoting “aging in place” [38]. These technologies incorporate digital health, which helps people manage and monitor their health [39]. Both older adults and their healthcare clinicians can use digital health technologies, including sensors found in wearable devices, smart homes, mobile and video communications, and social networks [36,37]. Telehealth, a video communication technology, allows healthcare providers to deliver medical care and education to people in remote rural areas [24]. These technologies may help improve health outcomes and increase access to nutrition information; nevertheless, use among older adults to provide routine nutrition care is limited [23,40].

Learned as an outcome of the COVID-19 pandemic, technologies can become a routine part of healthcare practice [41] and can support health management. In response, U.S. Medicare expanded coverage of telehealth services and agreed to temporarily reimburse RDNs for telephone assessments and services [42]. Some reimbursements continue as a new norm for coverage [43]. However, in the U.S., Medicare only covers medical nutrition therapy for older adults with diabetes, kidney disease, or those 36 months after a kidney transplant [23,40]. The pandemic has also reinforced older adults’ desire to “age in place” [38], which can be feasible with technology adoption.

### 1.5. Research Objective

Providing older adults with healthcare services using technology is efficient, convenient, and cost-effective. Technology reduces patient wait time, and research shows older adults find using technology to access healthcare services acceptable [24]. Exploring older adults’ technology use to manage their nutrition would be advantageous to better incorporate existing and new technologies into the older adults’ nutrition care plan. The primary objective of this scoping review is to assess the extant research on using technology to manage nutrition for and by older adults. The secondary objective is to describe the technology used and nutrition areas of focus of the extant research.

The remainder of this paper provides a detailed methodology of the search strategy, inclusion and exclusion criteria for studies, and how the articles were analyzed. The authors then describe the types of research studies extracted, the technology utilized, and the areas of nutrition the technology seeks to address. This paper closes with an interpretation and synthesis of the extracted studies, implication for practice, and recommendations for future research.

## 2. Materials and Methods

The Preferred Reporting Item for Systematic Reviews and Meta-analysis for Scoping Review (PRISMA-ScR) guidelines [44] and the PRISMA (http://www.prisma-statement.org/?AspxAutoDetectCookieSupport=1, accessed on 11 October 2023) (Preferred Reporting Items for Systematic Reviews and Metanalysis (PRISMA)) model [45] were used to perform a scoping review of the literature. Steps for conducting this review were adopted from the 5-step approach [46]. A scoping review methodology was selected based on the need to identify the type and extent of the research evidence [47], mainly to map out the type and nature of the technology used to manage older adults’ nutrition [48].

### 2.1. Research Methods

To build a foundation, we searched several nutritional science and business databases for peer-reviewed journal papers. At first, we assumed that if our research topic was an emerging phenomenon, research would exist in the top journals of our disciplines in nutrition and information systems (IS). The paper search began with prominent nutrition and IS journals. Table 1 summarizes the selected journals in alphabetical order. The IS journals chosen to be included in the preliminary search are considered prominent because they are included in the top eight journals as determined by the senior scholars of the Association for Information Systems [49]. Of the six nutrition journals selected, all but one are considered Quartile 1 or 2 journals based on the Clarivate Analytics Journal Citation Report [50]. The nutrition journals selected also frequently cover topics related to nutrition interventions, nutrition assessment, education, aging, and utilizing technology to manage nutrition. The one journal considered a Quartile 3 journal is the Journal of Nutrition in Gerontology and Geriatrics since the journal is dedicated to covering papers related to nutrition and aging.

Next, we researched prominent nutritional science and business databases. Our nutrition-focused databases were PubMed and Web of Science. The business databases for IS research were Business Source Elite, Abstracted Business Information/INFORM (ABI/INFORM) Collection, and ScienceDirect.

Finally, we considered secondary source papers. These papers surfaced through references within the papers retrieved from the databases or suggested by colleagues. While not comprehensive, this list represents a measure of older adult technology use of nutrition found in the nutrition, IS, and gerontology disciplines.

Our team of two authors and a trained research assistant searched for articles and checked each other’s work. We searched the databases with a combination of three groups of keywords: (1) “older adult”, “senior”, “elderly”, “geriatric”, “aging”, and “older person”; (2) “nutrition”, “diet quality”, “diet”, “nutritional status”, “health”, and “undernutrition”; and (3) “technology”, “telenutrition”, “telehealth”, “digital health”, “eHealth”, and “information systems”. The initial search and screening for articles occurred between January 2021 and March 2021.

### 2.2. Eligibility Criteria

Papers about older adults that addressed technology to manage nutrition were included. All papers were published between 1 January 2000 and 31 December 2020 and were written in English. Papers included qualitative and quantitative studies, mixed methods studies, randomized controlled trials, interventions, literature reviews, future research designs, and commentaries. Papers that did not include older adults or examine technology usage to manage nutrition were excluded. Additionally, we excluded studies of older adults that addressed only nutrition or technology, conference abstracts, and poster sessions.

### 2.3. Study Selection

We created a data extract for each database and journal search to record the citation and the paper’s PDF file. To understand how each study framed the search terms, we conducted an initial screening by examining the title, abstract, and keywords. We sought to understand whether our search word combination explained an older adult’s use of technology for nutrition care. Our abstract review examined how researchers referenced older adults. If terms such as elderly, seniors, and senior citizens were used, the paper was included. If the extant research included participants by age group and older adults were among them, the paper was included. We separately completed a full-text review of the retained articles against our inclusion criteria. We then met to discuss and resolve disagreements related to study inclusion. In a final evaluation of the remaining full-text articles, we excluded additional articles after further discussion.

### 2.4. Data Charting

We extracted the following information for each paper: authorship, type of study, study population, type of technology used, country, study purpose, and nutrition/medical outcomes. Data from each article were sorted into a summary table (Table 2, forthcoming) and synthesized. After that, the studies were characterized and described based on technology (Table 3) and nutrition (Table 4) areas of interest. Additionally, the two researchers further categorized the use of technology to manage and care for older adults’ nutritional needs into three domains—hardware, software, and various information and communications technology (ICT). For nutrition, the high-level categories were nutrition areas of focus and study setting.

Technology type data used in each study were recorded into the following categories: home-based sensors, smart devices (TV, smart scale, adaptive kitchenware), mobile device sensors, fitness devices, assistive robots, tablet computers, computers, smartphones, telephones, webcams (photos, video monitoring), internet access, videoconferencing (personal and health-related), and software (applications, web-specific resources) (Table 3). Additional information was extracted from each article to identify the technology used to manage nutrition, the nutrition focus, the specific end-users of interest, and the study setting (community vs. institutional) (Table 4). These categories were determined a priori and post hoc by the second author, an expert in IS, and finalized with the first author.

Data on each study’s nutrition area of focus were recorded and placed into the following categories and subcategories: assessment, monitoring, and/or tracking; weight and body composition; nutrition status; education/counseling; diet intake/diet quality; and activities of daily living (ADL). The first author, an RDN, identified these categories and finalized them with the second author. Each retained article was reviewed to understand who the end user was. We then noted the “distinction of end-user based on three types we created and defined as follows: (1) self-use of technology (SUT, where older adults use technology to manage nutrition and health); (2) provider use of technology to support older adults (PUT, where providers use technology to optimize or enhance the care of older adults); and (3) ability or readiness to use technology (ART, where older adults or providers received technology education or training, [51]”. Studies were characterized and described based on technology and nutrition areas of interest. After reviewing findings, integration of technology, and nutrition characterization, the two researchers further analyzed and integrated the nutrition and technology characterizations of interest into three domains.

**Table 2 nutrients-15-04402-t002:** Summary of studies using technology for managing nutrition for older adults.

ID	Authors	Type of Study	Population	Type of Technology	Location	Purpose	Medical/Nutrition Outcomes
S1	Ali et al., 2013 [52]	INT	Age 60–74 years, *n* = 6	Computer, software (3D animation enhancement to existing software for nutritional education) via an information kiosk	Kuala Lumpur, Malaysia	Investigate 3D animation software to develop nutritional education software using interface user design guidelines specific to older adults. Lack of computer literacy and impaired perception and cognition create a limitation in older adults’ use of computers. Assisted computer interaction with 3D animation may address that concern.	The 3D animation software helped facilitate older adults understanding of nutritional education content. Compared with a control group that used booklets, each group completed tasks, and the digital software user explored content using the software package faster than those using the traditional booklet.
S2	Angelini et al., 2016 [53]	Qual	Age 65 years and older aging at home, age 60 and older impending retirement, *n* = 100	Various types of information communication and technology: senior living lab, promoting aging at home with multidisciplinary co-creation of social and technology innovative products, services, and practices for older adults	Romandy, Switzerland	Co-creation of services and products for older adults adheres to an ecological approach to promote healthy aging. Using technology in the background to support healthy nutrition to cope with frailty, improved autonomous mobility to foster independence, and social communication of older adults. This study describes the roles technologies can have when co-creating with older adults, the methods needed, and the challenges.	This study discusses challenges that emerged during the first year of creating this new platform for investigating the role of information communication and technology (ICT) when designing products and services for older adults. Older adults are insecure with ICTs. The causes vary with age, culture, health conditions, and application context.
S3	Astell et al., 2014 [3]	Quant	Mean age 71.9 years (range 65–89), *n* = 40	Touch screen tablet computer, webcam, software (NANA)	Sheffield and St. Andrews, U.K.	Novel Assessment of Nutrition and Aging (NANA) validation study of older adults self-reported diet intake using a touch screen computer. The study also examined if data collection can occur over several nonconsecutive weeks. Bland–Altman analysis was used to explore differences between energy and macronutrient intake data.	The NANA toolkit was compared with standard measures of diet (four-day food diary), cognitive ability (processing speed), and physical activity. Data collected with the NANA toolkit were significantly correlated with a standard food diary, cognitive processing speed, and physical activity measures.
S4	Aure et al., 2020 [54]	Qual	Mean age 81 years (range, 68–95), *n* = 18	Tablet computer, software (APPetitus), mobile internet access	Three Norwegian municipalities	Undernutrition remains a prevalent and persistent problem among older adults. This study explored whether a tablet-based application supported nutritional self-care among older adults.	The app served as a reminder of available, relevant food options. By the end of the day, it encouraged some participants to eat or drink more, whereas others became aware of food selection options to ensure sufficient protein, energy, and fluids. Some participants made no effort to change their diet despite feedback that they did not eat or drink enough. Findings confirm the feasibility of using technology in nutrition interventions for older adults. Technical support from health professionals facilitated app and tablet use.
S5	Batsis et al., 2019 [55]	MM	Older adults, mean age 72.9 years (SD 4.6), *n* = 29; clinicians, mean age 47.7 years (SD 12.1), *n* = 7; community leaders, mean age 64.3 years (SD 8.73), *n* = 4; total *n* = 40	Fitness device, home-based sensors	New Hampshire, U.S.	Mobile Health Obesity Wellness Intervention (MOWI) in rural older adults with obesity, consisting of nutrition and exercise sessions, could be helpful to improve physical function and create accountability. The study purpose was to explore how technology could improve the health of rural older adults with obesity.	Older rural adults with obesity are interested in using technologies to improve their health. Barriers to implementation exist based on technology complexity and access. Clinicians and community leaders believed that technology could help provide older adults insight into health, create accountability, and motivate behavior change.
S6	Batsis et al., 2020 [56]	INT, Qual	Older adults, mean age 72.9 years (SD 4.6), *n* = 29; clinicians, mean age 47.7 years (SD 12.1), *n* = 7; community leaders, mean age 64.3 years (SD 8.73), *n* = 4; total *n* = 40	Tablet computer, software, telehealth via internet video devices, fitness device (obesity wellness)	New Hampshire, U.S.	Patient-oriented feedback with self-aware messaging based on adaptable sensor technologies was promising for eliciting behavioral change and could improve physical function and reduce obesity. This study sought to understand how a potential Mobile Health Obesity Wellness Intervention (MOWI) among older adults with obesity could enhance physical functioning. The remote intervention consisted of weekly nutrition counseling and biweekly exercise sessions.	There is potential acceptability and value for implementing a rural, telehealth-delivered intervention for older adults with obesity. Participants saw the potential of MOWI in overcoming geographic barriers to accessing healthcare for rural communities and creating accountability for participants. Participant feedback about improving implementation included adding regular social connectedness to the program.
S7	Beasley et al., 2019 [57]	INT, MM	Mean age 70.1 years (SD 5.6), *n* = 16	Fitness device, videoconferencing	New York City, U.S.	As of 2018, Medicare has covered the Diabetes Prevention Program (DPP), making it more accessible to older adults. This study tested the feasibility and acceptability of implementing a telehealth-adapted 6-week DPP at a New York City senior center. Feedback can be incorporated to design an effective trial.	Of the 16 participants recruited, retention was 75%, and the attendance rate averaged 80% across the six sessions. Focus groups provided positive opinions and suggested a greater focus on dietary strategies.
S8	Cabrita et al., 2019 [54]	Qual	Mean age 69 years (range 65–78), *n* = 12	Mobile phone, fitness device, smart scale, software (Activity Coach)	Overijssel, Netherlands	Low adherence to technologies to self-manage health among older adults may occur because their preferences are not considered when designing new health care technologies. This study examined older adults’ (1) current practices in health management; (2) attitudes toward technology to support healthy behaviors, including nutrition, physical and cognitive functioning, and well-being; (3) wishes and expectations from technology; and (4) attitudes toward using technology in health management after monitoring weight, physical functioning, and daily emotions for 1 month.	Participants saw an added value of using technology. Attitudes and wishes for technology to support health differed by health domain (i.e., nutrition, cognition, physical functioning, well-being). All participants saw the importance of keeping track of their diet, but 50% of participants would not use a website or an app to monitor diet. Participants were not aware of how technology could manage nutrition. Some wished they could receive healthy recipes tailored to medical needs. Fears around technology included: identity theft, replacement of human touch, and disuse of existing abilities. After 1 month of using technology, attitudes improved. Technology that supports aging in place should target health literacy.
S9	Chiu et al., 2019 [58]	INT, MM	Mean age 65.0 years (SD 8.33); group 1, age range 50–60 years, *n* = 6; group 2, age older than 65 years, *n* = 15; total *n* = 21	Touch screen tablet computer, internet, software (health knowledge)	Taiwan	Understanding whether mobile technologies support self-directed learning for older adults is unclear. This study assessed if nutrition education combined with mobile technology-supported teaching increases participant knowledge of and self-efficacy for a healthy diet.	Participants’ nutrition knowledge significantly improved; self-efficacy about a healthy diet showed marginal improvement. Nutrition knowledge was positively correlated with intensity of surfing the Internet or reviewing the electronic course material. Participants reported feeling “freshness”, “joyfulness”, and “great achievement” because of the combined course. Those who reviewed the electronic course material or searched for health information online showed a significantly greater understanding of and self-efficacy for a healthy diet.
S10	Dugas et al., 2018 [59]	Quant, INT	Mean age 67.6 years (SD 5.8), *n* = 27	Tablet computer, fitness device, software (DiaSocial)	Maryland, U.S.	MHealth tools’ effectiveness in managing chronic diseases, such as diabetes, warrants further exploration. Investigate the effectiveness of the mHealth app—intended to track glucose, exercise, nutrition, and medication adherence—for improving health behaviors among older veterans with poorly controlled type 2 diabetes using a 13-week pilot study.	Effectiveness of an intervention is conditional on locomotion, whereas eagerness to engage is motivated by goal app settings. Program adherence was associated with a more significant reduction in glycated hemoglobin (HbA1c) levels.
S11	Espín et al., 2016 [60]	FRD	Older adults	Computer, tablet computer, developed software (NutElCare)	Granada, Spain	This paper presents NutElCare, a nutritional recommendation system to help older adults develop healthy diet plans based on nutritional guidelines. This study highlights outcomes of nutritional recommender systems and the design and components of NutElCare; future directions are provided.	The research design for an app developed specifically for older adults provides recommendations based on expert guidelines for older adults to develop a diet plan. As an end-user influencer, the app includes adapting taste preferences.
S12	Farsjø et al., 2019 [23]	Qual	Group 1, older adults, mean age 78 years (range 41–96), *n* = 29; group 2, health care professionals, mean age 43 years (range 23–65), *n* = 24	Tablet computer, software (APPetitus), mobile internet access	Three Norwegian municipalities	Report the development of a nutrition application and the introduction of the app to healthcare professionals and older adults in-home care. Understand if healthcare professionals believe the apps are relevant and identify barriers to using the app for nutrition.	Access to technology enables older adults to take an active role in health monitoring. Goals for meeting enabled access include ease of use, support for the technology, and relevant app content.
S13	Göransson et al., 2020 [61]	INT	Mean age 86.0 years (SD 6.5), *n* = 17 older adults receiving home care	Smartphone, tablet computer, developed software (Interaktor)	Southwestern Sweden	Despite the increased use of mHealth tools in various populations, few studies have targeted older adults receiving home care services. Determine areas relevant to older adults’ health and self-care for regular assessment of support using a developed application for older adults receiving home care. Describe older adults’ usage of the app and evaluate the impact on health and health literacy over 6 months.	Findings show that older adults increased their communication and literacy knowledge through the application. This study highlights the importance of applications specifically designed for older adults. However, overall health among older adults did not improve at six months. Common self-reported health concerns that could impact nutrition include: difficulty performing activities of daily living, constipation, diarrhea, loss of appetite, difficulty eating.
S14	Hendrie et al., 2017 [62]	Quant	Group 1, age 18–30 years, *n* = 44,534; group 2, age 31–50 years; *n* = 52,599; group 3, age 51–70 years; *n* = 44,096; group 4, age 71+ years; total *n* = 145,975	Online survey (CSIRO healthy diet score survey)	Australia	This paper describes the research and user experience of the survey and summarizes how compliant self-reported diets are with the Australian dietary guidelines.	The development of an online survey estimates user compliance with dietary guidelines using an established healthy diet score. The study provided a one-time assessment in three areas where the survey user could improve. Although not statistically significant, adults 51–70 years and 70+ years had higher dietary scores than adults 50 years and younger.
S15	Hermann et al., 2012 [63]	Quant	Older adults, *n* = 100	Devices (assistive technology and appropriate software)	Oklahoma, U.S.	This study was performed to empower older adults with technology to shop, cook, and eat using an education program. Evaluate the effectiveness of a curriculum to increase awareness of assistive technology to manage food and nutrition.	Significant increases in awareness of the importance of nutrition, understanding, and likeliness of using information, and awareness and use of assistive technology devices. Likelihood of older adult contacting a government-sponsored assistive technology program increased.
S16	Kaddachi et al., 2018 [6]	Quant	Mean age 88.3 years (SD 4.5), *n* = 9	Inconspicuous technologies: Home-based sensors: environmental movement, door, bed, vibration, pressure, proximity, temperature, humidity, light. Smart devices: TV, scale, medicine box. Mobile device sensors (in-phone embedded): GPS, accelerometer, gyroscope, step detector, proximity	France	Identify significant behavior change indicators using statistical techniques that differentiate long-term and short-term changes in behavior. Early detection of long-term behavior changes in mobility, memory, nutrition, and social life indicators is important for improving older adults’ healthcare services.	Sensor data about the presence or absence of movements allows behavior change analysis, as seen in activity periods, room entries, sleep impairment, visits, time outdoors, and nutritional activities.
S17	Kirkpatrick et al., 2017 [64]	Quant	Four-part study: S1, age 2–5 years, *n* = 40; S2, age 10–13 years, *n* = 294; S3, age 10–13 years, *n* = 98; S4, age 36–82 years, *n* = 331; S5, age 48–88 years, *n* = 264	Various: smartphone, desktop, laptop, and tablet computer; internet; online survey (Automated Self-Administered 24 h) (ASA24)	Canada	Describe lessons learned from five studies that assessed the feasibility and validity of ASA24 for collecting dietary recall data among several population subgroups in Canada.	High acceptance of ASA24 was found among diverse samples. The ASA24 interface was not intuitive for young children and older adults; technological issues were encountered. The findings highlight the importance of piloting protocols and consideration of tailored resources to support participants. Older adults appeared to be more patient than younger adults by completing multiple passes to collect recall data.
S18	LaMonica et al., 2017 [65]	Quant	Mean age 67.6 years (SD 8.5), *n* = 221	Computer, smartphone, internet, website resources (health-related)	Sydney, Australia	The rapid increase in electronic health technologies warrants exploring whether these tools can be used for older adults with mild cognitive impairment. Describe patterns of Internet use and interest in and preferences for eHealth technologies among older adults with cognitive impairment. Prevalence data needed to determine the feasibility of future eHealth efforts for the aging population were collected.	Most participants used mobile phones (91.4%) and computers (86.1%) and had access to the Internet (92.6%). Preferences for other eHealth interventions varied with educational level; university-educated participants expressed greater interest in interventions related to mood (*p* = 0.01), socialization (*p* = 0.02), memory (*p* = 0.01), and computer-based exercises. eHealth preferences varied with the diagnosis for interventions targeting sleep, nutrition, vascular risk factors, and memory.
S19	Lete et al., 2020 [66]	Lit Rev	Older adults	Wearable devices (e.g., physical, affective, cognitive, clinical), sensors (e.g., physical, affective, cognitive, clinical, furniture, objects)	Spain	Virtual coaching is a promising option to help extend older adults’ time to live interdependently. Present a survey of different approaches in virtual coaching for older adults.	Coaching should be considered holistically, including physical and cognitive training, nutrition (self-management of weight and healthy eating behavior), social interaction, and mood.
S20	Lindhardt & Nielsen, 2017 [67]	MM, INT	Mean age 79.85 years (SD 7.85), *n* = 25	Tablet computer, developed software (nutrition application), internet	Denmark	Weight loss and low dietary intake during and after hospitalization are common among older adults and can impact health outcomes. Assess the acceptability, feasibility, and preliminary efficacy of technology-supported energy- and protein-enforced home-delivered meals for older adults discharged from the hospital.	Participants were motivated and capable of using the device; technology challenges were related to the immaturity of the out-of-hospital app version. Inclusion and retention were challenged by exhaustion among patients and the acceptability of the nutrition intervention; the mortality rate was high. The intervention group increased their muscle strength more consistently than the control group.
S21	Łukasik et al., 2018 [4]	MM	Group 1, older adults, mean age 75.3 years (SD 8.4), *n* = 126; group 2, caregivers, mean age 38.5 years (SD 13.0), *n* = 126; total *n* = 252	Assistive robot	France, Greece, Italy, Poland, Great Britain	Rapid development of new technologies has sparked interest in the use of assistive robots in managing the care of older adults. The study aimed to answer how both older people and caregivers perceive the possibility of using an assistive robot for nutritional support.	The diet of older adults was improved by advice on healthy eating or monitoring to improve diet. An age-related difference was observed. Older adults less frequently accepted reminders of mealtimes or drinking liquids than younger adults.
S22	Manea & Wac, 2020 [68]	Qual	Mean age 69.8 years (SD 7.4), *n* = 39	Fitness device	Hungary, Spain	This study showed the feasibility of a co-calibration method, coQoL, by quantifying relationships between patient-reported outcomes (PROs) and technology-reported outcomes (TechROs); PROs included nutrition, physical activity, social support, anxiety, depression, memory, quality of life, and sleep. The study assessed the quality of data collected from a wearable technology fitness device, while participants’ daily lives unfolded to inform the design of personalized behavioral studies.	High PROs and TechROs correlated with physical activity, social support, anxiety, and sleep of various durations. The coQoL method feasibly co-calibrates constructs within seniors’ physical behaviors and psychological states. Some PRO nutrition outcomes assessing Mediterranean dietary patterns and malnutrition had strong correlations with TechRO data.
S23	Marshall et al., 2017 [69]	Lit Rev	Age 65 years and older	Telehealth (nutrition care)	Australia	The demand for both domiciliary and family caregivers to provide in-home assistance for older adults, including food-related tasks, is increasing. A narrative review summarized the role of both domiciliary and family caregivers in providing individualized nutrition support for community-dwelling older adults with malnutrition.	Interventions reviewed including telehealth, group education, and skill development workshops show promise to improve outcomes of older adults. There is moderate evidence to support the inclusion of family caregivers as part of the nutrition care team. Moderate evidence supports the role of domiciliary caregivers in implementing nutrition screening and referrals and implementing malnutrition interventions with the support of health care professionals.
S24	Marx et al., 2018 [70]	Lit Rev, MM	Age 65 years and older	Telehealth via telephone or internet video devices (malnutrition-related care)	Brisbane, Australia; Oslo, Norway; Greifswald, Germany; “possibly Singapore or Malaysia;” Aarhus, Denmark; Herlev, Denmark; Netherlands	The effectiveness of telehealth to improve malnutrition among older adults requires exploration and understanding if health care resources may be appropriately engaged. Deliver malnutrition-related interventions to a group of community-dwelling older adults.	Malnutrition-related telehealth interventions for older adults living at home may improve quality of life and dietary intake. The approach seems feasible and cost-effective. Data suggest telehealth may improve nutrition status, physical function, hospital readmission, and mortality.
S25	McCabe et al., 2001 [24]	COM	Age 65 years and older	Telehealth via internet videoconferencing (medical nutrition therapy)	Arkansas, U.S.	Describe the use of telehealth for nutrition counseling of older adults in rural areas and the participation of dietitians in video technology for healthcare delivery.	Videoconferencing has provided rural hospitals with healthcare information and education for more than 6000 services to 45 rural facilities. Clinical consultations include access to information that may not have been otherwise available and access to specialists like dietitians for individual or group collaboration. A videoconferencing nutrition consultation is described, as is the role of the caregiver during these virtual nutrition consultations.
S26	McCauley et al., 2019 [71]	COM	Age 65 years and older	Software (malnutrition quality improvement integrated into electronic health care records)	U.S.	The purpose of the Malnutrition Quality Improvement Initiative (MQii) is to: (1) serve as a toolkit used by an interdisciplinary team to improve the effectiveness and timeliness of malnutrition care; (2) facilitate the adoption of malnutrition electronic clinical quality measures (eCQMs) to help improve health outcomes; and (3) expand the availability of tools that can be integrated into electronic health record (HER) systems to minimize administrative burden and improve quality of care and documentation.	Innovations included the development of electronic clinical measures of malnutrition, such as global composite measures and a complementary interdisciplinary quality improvement toolkit. The initiative established the first nutrition-focused national learning collaborative.
S27	Moguel et al., 2019 [72]	Lit Rev	Age 65 years and older	Food intake monitoring technology solutions: Smartphone. Image-based (webcam). Wearable device. Smart home (sensor). Internet of things. Tablet computer.	Spain	Studying the extant literature is necessary to understand if technology is a viable solution for older adults in rural settings. Evaluate the suitability of food intake monitoring systems for older adults in rural regions and existing technological proposals for food intake monitoring.	A complete solution for monitoring the diet of older adults in a rural setting does not exist. Future efforts should include technology that identifies the user, addresses a solution for uploading user data, enhances self-service treatment solutions, and incorporates self-adaptive demographic profiling solutions to bridge the gap between users and technology knowledge.
S28	Ploeg et al., 2019 [73]	Qual	Mean age 78.7 years (SD 6.1), *n* = 32	Tablet computer, software (collection of information health risk, needs, and goals); personal health records and secure messaging	Ontario, Canada	Person-centered health care warrants that patient care is directed toward capable people based on their preferences, needs, and values. Understand how a new multicomponent care program improves the quality of primary care.	The program was a valuable tool for assessments, seminars, and an interdisciplinary approach to care. How the information was shared, and the kind of benefit that could be expected were unclear.
S29	Pownall et al., 2019 [74]	Qual	Care home staff, nursing, care assistant team, and catering staff, *n* = 37; care home managers, *n* = 4; quality managers, *n* = 4; residents, *n* = 6	Tablet computer, software (education and monitoring application for staff for patient dysphagia)	England	Dysphagia (difficulty or discomfort in swallowing) is a risk factor for poor nutrition among older adults, but optimizing support for nursing home residents can be challenging. Evaluate use of a digital dysphagia guide in care homes. Data are based on a consensus from interviews and focus groups on prioritizing the need for information and exploring the acceptance of an education tool for care workers.	A tablet-based digital guide to dysphagia provides care homes with an applied, interactive, work-based approach to education and training of the entire workforce. Videos, text, and photos were valuable for addressing different learning styles. A resource accessible in snips of learning addresses a range of learning styles. The flexibility of computer-based content in real-time enhances knowledge and skill development for caregivers.
S30	Qian & Gui, 2020 [75]	Quant	Age 60 years and older, *n* = 14,933; website posts	Website resource (development of senior online communities) (SOCs)	Wuhan, China	This study identified the health information needs of SOC users to help improve health information services for older adults. Three research questions were asked: (1) What type of health information is discussed in SOCs? (2) Based on the health information that users post in SOCs, do their health information needs change over time (including preference for traditional Chinese medicine (TCM) vs. Western medicine)? (3) How popular are different types of health information posted in SOCs?	Four types of health information were provided: “coping with aging, dietary nutrition, physical exercise, and mental health”. Older users reported comprehensive needs that involve various health issues, with the main concern being physical health. A larger number of posts were related to Western medicine than to TCM. Posts related to TCM mainly were associated with the categories “coping with aging” and “physical exercise”, whereas the proportion of “dietary nutrition” posts related to TCM was lower. Related to the category “dietary nutrition”, terms that came up in SOCs were “vitamins, fats, and proteins”, indicating that older adults may focus on problems with a healthy diet.
S31	Recio-Rodríguez et al., 2019 [76]	FRD	Age 65–80 years, *n* = 160	Smartphone, fitness device, software (activity and diet (Intellectual Property Registry No. 00/2017/2438)).	Spain	Evaluate the effectiveness of the combined use of smartphone and smart band technology for three months combined with counseling (intervention) vs. counseling alone (control) to increase physical activity levels and adherence to the Mediterranean diet. Assess the effect of the two interventions on body composition, cognitive performance, quality of life, and independence in activities of daily living.	The intervention group was instructed on how to use a smartphone application for three months. The application integrates physical activity information from a fitness bracelet and self-reported daily nutritional intake. Outcomes measured included: change in the number of steps measured with an accelerometer, adherence to the Mediterranean diet, sitting time, body composition, quality of life, cognitive performance, and independence in activities of daily living. All variables were measured at baseline and after three months. At six months, a follow-up telephone call collected dietary and physical activity data.
S32	Roberts et al., 2020 [77]	Qual	Age 60–83 years, *n* = 11	Touch screen tablet computer, software (ordering meals, self-monitored dietary intake, and guided nutrition goal setting)	Australia	This research was a sub-study of a feasibility study, with the primary aim to explore patient perceptions and acceptability of a health information technology intervention to improve the dietary intake of patients during hospitalization.	Two main themes emerged. The first theme captured experiences and perceptions of using technology to participate in nutrition care. Patients found the technology useful, valuable, and easy to use, but they valued interactions with staff. The second theme captured the spectrum of patient participation, ranging from learning about nutrition to self-monitoring and evaluating and showing behavior change. Patients enjoyed gaining nutrition awareness and knowledge. Most patients self-monitored food intake and goals being assessed, and some reported changing the foods they ordered based on the information learned.
S33	Scott et al., 2018 [78]	Lit Rev	Older adults	Emerging technology: Robot. Smart home. Internet of things. Sensor. Three-dimensional food printer. Technology for activity: Wearable. Software. Video game. Virtual reality. Technology for nutrition: Software. Meal delivery.	Australia	This study investigated the roles of assistive technology to overcome sarcopenia-related functional decline. Management components included poor strength and mobility and/or supporting health behaviors, including nutrition, which can help prevent sarcopenia progression.	There is limited evidence of the role of assistive technology for persons with sarcopenia. Promising areas for this population include smartphone applications, smart homes, wearables, robotics, and 3D food printers. Assistive technologies may contribute to maintaining adequate nutrition and physical activity, which may slow the condition’s progression.
S34	Sheats et al., 2017 [79]	MM	Mean age 70.8 years (SD 7.7), *n* = 23	Tablet computer, software (Discovery tool, application to collect data about aspects of the environment that may facilitate or hinder healthy living)	California, U.S.	The aging US population warrants a better understanding of ecological factors and their impact on older adults’ food environment. The goals of the Food Environment Assessment Study (FEAST) were to: (1) use the Healthy Neighborhood Discovery tool (Discovery tool) to collect data (geocoded photos, audio narratives) about aspects of the participants’ environmental facilitators and barriers to healthy living; food-related behaviors were also assessed and (2) use the findings to advocate for change in partnership with local decision-makers and policymakers.	Access to affordable, healthy food and transportation provided a significant barrier to healthy eating and navigating the local food environment. Participants were trained in advocacy skills and shared findings with relevant policymakers. At months 3, 6, 12, and 24, proximal and distal effects of the community-engagement process were documented and showed individual-, community-, and policy-level impacts. Findings add to the literature on how low-income, racially diverse, older adults are impacted at the individual, social, and environmental levels to access, choose, and purchase healthy foods. Multilevel solutions involving a variety of sectors are needed.
S35	Singer et al., 2018 [80]	INT	Mean age 62.9 years (SD 5.7), *n* = 15	Tablet computer, smartphone, fitness device, software (enhancement to existing software (Aidcube^TM^ (www.aidcube.com)) for home exercise, diet, and dietary goals), telephone	San Francisco, California, U.S.	A pilot study to demonstrate feasibility and safety for improving frailty using a targeted two-phase intervention: (1) assessment, training, and baseline exercise prescription and (2) home-based exercise and nutrition recorded with software.	A home-based exercise and nutrition intervention is feasible, safe, and capable of improving frailty in adult candidates for a lung transplant. More than half of the participants improved their frailty scores by the minimum clinically important difference, and an equal number of participants went from frail to not frail.
S36	Takemoto et al., 2018 [81]	COM	Age 65 years and older	Novel hardware and software technology (i.e., mobile devices, sensors, home-based sensors, software (monitoring, games), smartphones, webcams, tablets, telephone)	U.S.	Research focused on how novel technologies are applied with older adults and the many barriers when introducing technology within this demographic. Many technology applications are available to monitor health events and behaviors to aid with the changes associated with aging.	Findings on older adult lifestyle behavior and technology show that research is early and often found only in small pilot studies. More extensive trials are needed to understand better techniques and limitations to address the design of computer-based interventions using technology to assist with and easily collect accurate information about day-to-day quality of life.
S37	Timon et al., 2015 [82]	INT	Study 1, mean age 72.0 years, *n* = 40; study 2, mean age 75.9 years, *n* = 18; study 3, mean age 71.8 years, *n* = 36; total *n* = 94	Touch screen computer, software (record dietary intake and photo of food), and webcam	Sheffield and York, England	Many computer-based dietary assessment methods have been designed for children and the general adult population. There is a need for such an appropriately designed technology for older adults. Assess the validity of the Novel Assessment of Nutrition and Ageing (NANA) method for dietary recording and assessment in older adults using a touch-screen computer and webcam to understand an appropriate method for tracking nutrition among older adults.	Results indicate that the NANA method is appropriate for assessing dietary intake in older adults. The NANA method compares well with a four-day estimated food diary; potential technology-based food diary intake record for older adults.
S38	van den Helder et al., 2018 [83]	FRD, MM	Age 55 years and older, *n* = 240	Tablet computer, software (exercise application)	Amsterdam, Netherlands	An application was developed for a home-based vitamin program using a tablet computer. The study was based on content specific to safe exercise and vitamin and protein intake for older adults. It aimed to understand the impact of home-based exercise and protein intake through a technology tool for community-dwelling older adults. This paper describes the study protocol.	This is the first study to investigate the impact of home-based exercise, protein intake and technology for this population.
S39	van Doorn-van Atten et al., 2019 [84]	INT, MM	Mean age 77.3 years (SD 7.2), *n* = 76 completers	Tablet computer, computer, software (diet and nutritional questionnaires)	Netherlands	Undernutrition can negatively impact health and quality of life in older adults. Better monitoring of community-dwelling older adults’ nutritional status is needed. Conduct a process evaluation of a multicomponent nutritional telemonitoring intervention.	Eighty percent of participants completed the intervention. Non-completers were older and had worse physical and cognitive functioning. There was better adherence to weight telemonitoring than to telemonitoring using questionnaires. The intervention was well-received by older adults, with high satisfaction; satisfaction was lower among nurses.
S40	van Doorn-van Atten et al., 2018 [85]	INT, FRD	Age 65 years and older, *n* = 215	Computer, tablet computer, Bluetooth, TV (Bluetooth-capable), software (diet and nutritional questionnaires)	Netherlands	Older adults are at risk for malnutrition, and there is a need for innovative resources to monitor and improve nutritional status. Describe an intervention study design that uses telemonitoring to improve the nutritional status of community-dwelling older adults.	The six-month intervention was evaluated using a parallel arm pretest. The intervention group received nutritional telemonitoring, television messages, and dietary advice from a nurse or a dietitian. The control group received usual care. Measurements collected at baseline, after 4.5 months, and at completion. Measures included: nutritional status, behavioral determinants, diet quality, appetite, body weight, physical activity, physical functioning, and quality of life. A process evaluation assessed delivery, feasibility, and acceptability.
S41	van Doorn-van Atten et al., 2019 [40]	MM	Mean age 77.4 years (SD 9.3), *n* = 11	Computer, tablet computer, TV, TV set-top box, internet, software	Netherlands	Evaluate the feasibility and effectiveness of a three-month telemonitoring intervention to improve community-dwelling older adults’ nutritional status and health outcomes.	This intervention was implemented by researchers and healthcare professionals as intended. Healthcare professionals found the intervention acceptable. Of 20 participants, 9 dropped out. Participant acceptance was low due to low usability of the telemonitoring television channel. Participants had good adherence but needed more help than anticipated with using technology. Compliance with several dietary guidelines was observed. No effects on nutritional status, physical functioning, or quality of life were found.
S42	Ventura Marra et al., 2019 [86]	INT	Intervention group, mean age 58.6 years (SD 8.1), *n* = 29; enhanced usual care group, mean age 59.3 years (SD 7.4), *n* = 30; total *n* = 59	Telehealth via internet videoconferencing and telephone (registered dietitian nutritionist)	U.S.	Overweight and obesity negatively impact health status, functionality, and quality of life of adults and older adults. Weight loss interventions have been predominantly conducted among women, and accessing nutrition services in rural communities can be challenging. Evaluate the feasibility and effectiveness of a 12-week primary care-referred telenutrition weight loss intervention.	Both groups saw a significant reduction in body weight, waist circumference, body fat percentage, calorie intake, and improved diet quality. After controlling for time, no difference was seen between groups. At 12 weeks, a more significant proportion of participants in the intervention group had lost at least 5% of their baseline weight than the enhanced usual care group. Retention rates and participant-reported satisfaction and adherence were greater than 80% in the telenutrition group. A larger trial over a more extended period is warranted.
S43	Ward et al., 2019 [87]	MM	Age 67–77 years, *n* = 282	Computer, tablet computer, internet, software (Myfood24, online 24 h dietary recall application, (www.myfood24.og))	U.K.	Collecting accurate dietary assessment data is essential to understand diet–disease associations at different stages of life. Evaluate the feasibility of using Myfood24 as a dietary assessment tool for older adults.	Overall, 67% of participants completed at least one recall, and 48% completed two or more. Participants who completed multiple recalls reported higher self-confidence with technology and received a higher technology readiness score than those who did not complete any recalls. Additional support may be required to obtain multiple dietary recalls in an older adult population.
S44	West et al., 2010 [88]	RCT	Age 55–64 years, 14.4%; age 65–69 years, 32.3%; age 70–74 years, 24.4%; age 75–79 years, 16.6%; age 80 years and older 12.3%; total *n* = 610	Telehealth via internet videoconferencing (certified diabetes educator)	Rural upstate New York, U.S.	Understand the use of telemedicine for setting goals to address behavior change; examine progress toward these goals in underserved rural older adults with diabetes.	Telemedicine is an acceptable tool for consistent diabetes education, nutrition counseling, and diabetes monitoring. It offers access to diabetes support, resources, and feedback from the convenience of their home. Overall, 68% of behavioral goals set during the intervention were rated as “met” or “improved”.

Table Legend: App = application; COM = commentary; FRD = future research design; INT = intervention; Lit Rev = literature review; MM = mixed methods; Qual = qualitative; Quant = quantitative; RCT = randomized controlled trial.

**Table 3 nutrients-15-04402-t003:** Forms of technology used to manage nutrition for older adults.

			Hardware										Software			
ID	Study	End-User	Home-Based Sensors	Smart Devices	Mobile Device Sensors	Fitness Devices	Robots or Assistive Robots	Tablet Computer	Desktop or Laptop Computer	Mobile or Smartphone	Telephone	Webcam	Internet Access	Videoconferencing	Applications	Various ICT
S1	Ali et al., 2013 [52]	ART							√						√	
S2	Angelini et al., 2016 [53]	SUT PUT														√
S3	Astell et al., 2014 [3]	SUT						√				√			√	
S4	Aure et al., 2020 [54]	SUT						√					√		√	
S5	Batsis et al., 2019 [55]	SUT PUT	√			√										
S6	Batsis et al., 2020 [56]	SUT PUT				√		√						√	√	
S7	Beasley et al., 2019 [57]	SUT				√								√		
S8	Cabrita et al., 2019 [89]	SUT		√		√				√					√	
S9	Chiu et al., 2019 [58]	SUT						√					√		√	
S10	Dugas et al., 2018 [59]	SUT				√		√							√	
S11	Espín et al., 2016 [60]	SUT						√	√						√	
S12	Farsjø et al., 2019 [23]	SUT PUT						√					√		√	
S13	Göransson et al., 2020 [61]	SUT						√		√					√	
S14	Hendrie et al., 2017 [62]	ART													√	
S15	Hermann et al., 2012 [63]	ART		√											√	
S16	Kaddachi et al., 2018 [6]	PUT	√	√	√											
S17	Kirkpatrick et al., 2017 [64]	PUT						√	√	√			√		√	
S18	LaMonica et al., 2017 [65]	ART							√	√			√		√	
S19	Lete et al., 2020 [66]	PUT	√		√											
S20	Lindhardt & Nielsen, 2017 [67]	SUT PUT						√					√		√	
S21	Łukasik et al., 2018 [4]	SUT, PUT					√									
S22	Manea & Wac, 2020 [68]	SUT PUT				√										
S23	Marshall et al., 2017 [69]	PUT												√		
S24	Marx et al., 2018 [70]	SUT PUT									√			√		
S25	McCabe et al., 2001 [24]	SUT PUT												√		
S26	McCauley et al., 2019 [71]	PUT													√	
S27	Moguel et al., 2019 [72]	SUT PUT	√		√			√		√		√	√			
S28	Ploeg et al., 2019 [73]	PUT SUT						√							√	
S29	Pownall et al., 2019 [74]	PUT						√							√	
S30	Qian & Gui, 2020 [75]	SUT													√	
S31	Recio-Rodríguez et al., 2019 [76]	SUT				√				√					√	
S32	Roberts et al., 2020 [77]	SUT PUT						√							√	
S33	Scott et al., 2018 [78]	SUT	√	√	√		√						√		√	√
S34	Sheats et al., 2017 [79]	SUT						√							√	
S35	Singer et al., 2018 [80]	SUT PUT				√		√		√	√				√	
S36	Takemoto et al., 2018 [81]	ART	√		√			√		√	√	√			√	√
S37	Timon et al., 2015 [82]	SUT						√				√			√	
S38	van den Helder et al., 2018 [83]	SUT						√							√	
S39	van Doorn-van Atten et al., 2019 [84]	SUT						√	√						√	
S40	van Doorn-van Atten et al., 2018 [85]	SUT		√				√	√						√	√
S41	van Doorn-van Atten et al., 2019 [40]	SUT		√				√	√				√		√	
S42	Ventura Marra et al., 2019 [86]	SUT									√			√		
S43	Ward et al., 2019 [87]	SUT						√	√				√		√	
S44	West et al., 2010 [88]	SUT												√		
Total:			6	6	5	8	2	23	8	8	4	4	10	7	31	4

Table Legend: ART = ability or readiness to use technology; ICT = information and communication technology; PUT = provider use of technology in support; SUT = self-use technology [51].

**Table 4 nutrients-15-04402-t004:** Nutrition area of focus and study setting to manage nutrition for older adults.

		Nutrition Area of Focus						Study Setting	
ID	Study	Assessment, Monitoring, and/or Tracking	Weight and Body Composition	Nutrition Status	Education/ Counseling	Diet Intake/ Diet Quality	Activities of Daily Living	Acute/Long-Term Care	Community Dwelling
S1	Ali et al., 2019 [52]				√				√
S2	Angelini et al., 2016 [53]	√	√				√ frailty		√
S3	Astell et al., 2014 [3]	√ detect for poor nutrition status to prevent frailty/sarcopenia				√			√
S4	Aure et al., 2020 [54]			√	√				√
S5	Batsis et al., 2019 [55]		√						√
S6	Batsis et al., 2020 [56]	√	√		√				√
S7	Beasley et al., 2019 [59]		√		√	√			√
S8	Cabrita et al., 2019 [89]	√							√
S9	Chiu et al., 2019 [58]				√				√
S10	Dugas et al., 2018 [59]	√			√	√			√
S11	Espín et al., 2018 [60]	√		√	√	√			√
S12	Farsjø et al., 2019 [23]	√	√	√	√	√			√
S13	Göransson et al., 2020 [61]	√		√	√				√
S14	Hendrie et al., 2017 [62]					√			√
S15	Hermann et al., 2012 [63]				√		√		√
S16	Kaddachi et al., 2018 [6]	√					√		√
S17	Kirkpatrick et al., 2017 [64]	√				√			√
S18	LaMonica et al., 2017 [65]	√			√				√
S19	Lete et al., 2020 [66]	√			√		√		√
S20	Lindhardt & Nielsen, 2017 [67]	√	√	√	√	√			√
S21	Łukasik et al., 2018 [4]	√			√	√	√		√
S22	Manea & Wac, 2020 [68]	√							√
S23	Marshall et al., 2017 [69]			√		√			√
S24	Marx et al., 2018 [70]			√					√
S25	McCabe et al., 2001 [24]	√			√				√
S26	McCauley et al., 2019 [71]			√				√	
S27	Moguel et al., 2019 [72]								√
S28	Ploeg et al., 2019 [73]	√			√				√
S29	Pownall et al., 2019 [74]				√	√ hydration		√	
S30	Qian & Gui, 2020 [75]					√			√
S31	Recio-Rodríguez et al., 2019 [76]				√	√			√
S32	Roberts et al., 2020 [77]	√ screening		√	√	√		√	
S33	Scott et al., 2018 [78]	√	√ sarcopenia			√	√		√
S34	Sheats et al., 2017 [79]	√environment							√
S35	Singer et al., 2018 [80]		√ frailty		√				√
S36	Takemoto et al., 2018 [81]	√	√	√	√	√ hydration			√
S37	Timon et al., 2015 [82]	√				√			√
S38	van den Helder et al., 2018 [83]		√		√	√			√
S39	van Doorn-van Atten et al., 2019 [84]	√		√	√	√			√
S40	van Doorn-van Atten et al., 2018 [85]	√	√	√	√	√			√
S41	van Doorn-van Atten et al., 2019 [40]	√	√	√	√	√			√
S42	Ventura Marra et al., 2019 [86]		√		√	√			√
S43	Ward et al., 2019 [87]	√							√
S44	West et al., 2010 [88]	√			√	√			√
Total:		27	13	13	27	23	6	3	41

Table Legend: Acute = acute care, hospital; ADL = activities of daily living, such as preparing food; LTC = long-term care facilities (e.g., nursing home, assisted living facility) [51].

## 3. Results

From the initial search, 254 papers were extracted, and 52 were added as identified with other sources for 306 papers. A review of the remaining full-text articles was completed, where additional articles were excluded from this study based on a lack of discussion of nutrition, technology, or aging, resulting in the removal of 165 papers. Duplicates were removed from the 141, leaving the full-text analysis of 79 papers. For the first round of analysis, we removed 25 papers since they did not fully cover the topic areas. In a second round of review, 11 papers were removed as a meticulous final review required that papers be excluded. Debate and agreement on the inclusion or exclusion of a paper ensued between the authors to develop the final list of papers. Forty-four full-text peer-reviewed papers met the research criteria and were retained, as shown in Figure 1: process of identification, and inclusion in this study: PRISMA diagram flow.

### 3.1. Study Characterization

The final papers fit within eight categories: (1) commentary, (2) future research, (3) intervention, (4) randomized, (5) literature review, (6) qualitative, (7) mixed methods, and (8) quantitative. Some retained papers fit into multiple study categories, yet only the first category associated with the study was summarized. Most studies examined the early stages of using technology to manage nutrition, including older adults’ and providers’ technology preferences, research protocols, and pilot/feasibility interventions. Mixed methods and qualitative studies explored the end users’ experiences and the cocreation of digital nutrition interventions. Quantitative studies compared digital technologies with traditional methods for collecting nutrition data [3,6,64,65].

### 3.2. Participant Characterization

Most studies’ target population focused on older adults (Table 2). Three studies included older adults and their caregivers [4,55,56]. One study examined home care staff working with older adults [74]. Three studies included older adults and other age groups [53,62,64]. Eight studies included adults 50 years and older [52,58,62,75,80,83,86,88]. Older adults of interest included those with elevated body mass index [55,56,86]; those who were lung transplant candidates [80]; and those with diabetes [59,88], prediabetes [57], and cognitive impairment [64]. Most studies addressed community-dwelling older adults (*n* = 41). Target end users included: 23 papers on SUT (51.1%), 7 on PUT (15.6%), 11 on SUT and PUT (24.4%), and four on ART (8.9%) as shown in Figure 2: target end users of technology.

### 3.3. Technology Characterization

Our results revealed that most studies included multiple technologies, particularly a combination of hardware and software (Table 3).

Software applications: Most studies extracted utilized software applications, specifically to “assess, track and monitor” older adults’ health outcomes, including dietary intake (*n* = 31 [3,23,40,52,54,56,58,59,60,61,62,63,64,65,67,71,73,74,75,76,77,78,79,80,81,82,83,84,85,87,89]). Thirteen applications served to help provide food options, recommendations, remind older adults to eat or drink, or provide nutrition education [23,40,52,54,56,58,60,61,67,75,77,83,85]. Nine studies used a software application to track and analyze diet [3,59,62,64,76,79,82,87,89]. Timon et al. [82] validated the Novel Assessment of Nutrition and Aging (NANA) method, where participants recorded their food intake in software and took photos of the food consumed. Two software applications, the Malnutrition Quality Improvement Initiative (MQii [71]) and a digital Dysphagia Guide with Care Homes [74] were created to assist healthcare professionals in improving the care provided. The MQii [71], along with the Health Tapestry program (TAP-App [73]), integrated their applications with electronic health records to improve the quality of care provided and documentation.

Software applications were accessed using a tablet computer (*n* = 21 studies, [3,23,40,54,55,58,59,60,64,67,73,74,75,77,79,80,81,82,83,84,85,87]) or mobile/smartphone devices (*n* = 7 studies [61,64,65,76,80,81,89]). Nine extracted studies that utilized software applications also incorporated internet access, where the end users needed the internet to connect to digital healthcare resources or providers [6,23,54,58,64,67,78,84,89]. Three studies using software applications utilized webcam technologies [3,81,82]. In addition to utilizing the software application, some studies used other devices or sensors: five studies utilized a fitness device along with software [56,59,76,80,89]. Three studies utilized a smart device with a software application [40,63,89]. Takemoto et al. [81] described how technology could enhance diet and activity assessments, demonstrating the use of software and home-based and mobile device sensors. Scott et al. [78] reviewed how assistive technologies, such as robots, home-based sensors, and mobile device sensors, could help overcome sarcopenia in aging, which included a description of software applications.

Tablet computers: The second highest-use area of technology that emerged in the literature is hardware. Significantly, tablet computer personal devices are most prominent for software use and internet access. The most common technological hardware device utilized in the extracted studies was tablet computers (*n* = 23). Most of the studies used tablets for participants to access software [3,6,23,40,55,58,59,60,61,64,67,73,77,78,79,80,81,82,83,84,87]. Three studies that used a tablet computer also used a fitness device [55,59,80]. Four studies utilized tablet and webcam technology [3,72,81,82]. Some studies looked at different devices to provide or access nutrition services, such as tablet computers, smartphones, desktops, or laptops [40,60,61,64,72,80,81,84,85,87]. Tablet use was for various nutrition areas of focus, such as collecting and monitoring nutrition outcome measures, including dietary intake and providing nutrition education.

Notable mention technologies: To a lesser extent, the following technological devices or software were found in the analysis: sensors and smart devices, robots/assistive devices, desktop/laptop, mobile or smartphone, webcam, telephone, video conferencing, and internet access. Nineteen studies explored the utilization of sensors and smart devices; this included home-based sensors (*n* = 6 studies), mobile sensors (*n* = 5 studies), and fitness devices (*n* = 8 studies). Ten studies utilize internet access as a core component of delivering nutrition or health information.

Integration of technology and nutrition characterization: The use of technology is the integration of “software and hardware [in] three domains as it relates to nutrition management and the nutrition care process: (1) “track, plan, and execute” (track dietary intake, plan changes, and execute nutrition plans); (2) “assess” (health data collection); and (3) build knowledge (promote clinicians’ and older adults’ nutrition understanding [51]).

### 3.4. Nutrition Focus Characterization

Most studies focused on multiple nutrition areas (Table 3), including assessment, monitoring, and/or tracking [3,4,6,23,24,53,56,59,60,61,64,65,66,67,68,73,77,78,79,81,82,84,85,87,88,89], nutrition education and counseling [4,23,24,28,40,52,54,55,56,58,59,60,61,63,65,66,67,73,74,75,76,80,81,83,84,85,86,88], and dietary intake and quality [3,4,23,40,57,59,60,62,64,67,69,74,75,76,77,78,81,82,83,84,85,86].

Nutrition assessment and monitoring: Many studies focused on nutrition assessment, monitoring, or tracking dietary behaviors. In some studies, self-monitoring was assessed, where older adults tracked dietary intake and nutrition-related markers, such as weight and laboratory values [3,23,40,59,61,64,65,66,67,73,77,82,85,89]. Other studies focused on passive monitoring, where technology was used to monitor and track older adults’ movement and behaviors using wearable devices, sensors, and robots [4,6,53,60,68,81]. In several studies, health professionals performed assessments and monitoring with videoconferencing [24,56,87,88]. One study had older adults use a tablet application to collect data, these data were used to identify environmental facilitators and food barriers to healthy living [79].

Nutrition education and counseling: Studies that focused on nutrition education included remote nutrition counseling and self-directed education, along with articles that outline how technology can be used to improve dietary intake and nutritional status. Several studies focused on webinars and “live” nutrition counseling [57,76,83,88] as ways to provide nutrition information. Two papers used telephone sessions with an RDN [8,80]. One study trained long-term care staff to enhance their knowledge around nutrition related topics [73]. One study educated providers on enhanced caregiver knowledge of dysphagia in long-term care settings [73].

Studies that focused on self-directed nutrition education provided older adults nutrition information. Two studies blended education sessions and self-directed learning [58,73]. Studies on self-directed nutrition education typically had older adults receive nutrition information, such as recommendations, recipes, or patient-oriented feedback and messaging. Afterward, older adults set personal nutrition goals to improve self-care, usually using an application [23,40,52,54,56,59,60,61,67,77,84,85]. Most studies focused on self-directed nutrition education were pilot studies, protocol papers, or formative studies that examined older adults’ preferences and acceptability after interacting with the application. Chiu et al. [58] and Ploeg et al. [73] utilized a combination of live-nutrition education sessions and self-directed learning. Participants attended nutrition lectures and received a device with nutrition-related films and applications downloaded [58]. Ploeg et al. [73] conducted home visits, where older adults set goals with their healthcare providers. The health data were stored on personal health records that both older adults and their providers could access, and older adults could message their providers using their personal health records.

Some studies examined older adults’ preferences to better understand how to provide nutrition information [4,70,71]. One study found that older adults and their caregivers thought it would be helpful if a robot provided nutrition advice. Two studies synthesized the literature and provided the best nutrition and health education practices using technology [24,81].

Diet intake and quality: About half of the studies focused on assessing and tracking dietary intake or improving diet quality. Studies examined how technology can be used to evaluate dietary intake [3,23,57,82], diet quality [40,84,85], or dietary pattern adherence [59,62,64,76,86].

Other studies provided tailored meal plans to improve nutrition [60,67,77]. Two papers focused on how caregivers could improve older adults’ dietary intake [69,74]. Other studies obtained older adults’ views on how technology could improve their nutrition [4] or synthesized the previous literature [78,81]. For example, Lukasik et al. [4] collected formative data on how robots may help offer nutrition support to older adults and caregivers. Their findings reveal that older adults accepted that a robot could remind them to eat and drink for optimum food and fluid intake. Finally, one study looked at older adults who participated in online communities and found that older adults’ posts related to “dietary nutrition” often revolved around certain dietary nutrients of concern, such as vitamins, fats, and protein [75].

Other nutrition areas: Less frequently considered nutrition topics included: nutritional status (*n* = 12 studies), weight and body composition (*n* = 12 studies), and ADL (*n* = 7 studies). Several studies used technology to improve nutrition status for those older adults that presented with malnutrition or that were at nutritional risk [40,54,67,84,85]; these studies explicitly addressed frailty and sarcopenia and optimizing weight status among older adults. Other studies assessed malnutrition but without technology [60,61,77]. Three studies examined how technology can promote weight loss [55,56,86]. One study described how caregivers could help treat and prevent protein-energy malnutrition. Two studies examined the existing research on how technology can help prevent and manage malnutrition [70,81]. Specifically, Marx et al. [70] completed a systematic review assessing how malnutrition-focused telehealth interventions could improve protein intake among older adults and QOL. Several studies addressed how technology can help older adults maintain their independence by completing food-related ADLs, including buying, preparing, and eating food safely and reminding them to consume meals and snacks.

## 4. Discussion

This scoping review examined the existing technologies used to support older adults in managing their nutrition and food intake. It also described the areas of the NCP in which technology can better serve older adults. Software applications were the most common form of software mentioned in the literature. Tablet computers were the most used hardware device. Figure 3, prominent uses of software application and tablet computers, summarizes the areas of technology used to manage nutrition by software applications and tablet computers. The primary nutrition areas of focus included: nutrition assessment, monitoring, and/or tracking, nutrition education/counseling, and diet intake and quality. These categories are integral pieces of the four steps of the NCP. Many of the nutrition interventions sought to address age-related nutrition concerns that are well-documented among this population and include: malnutrition, sarcopenia, frailty, and chronic conditions with dietary implications. Additionally, the findings reveal that while telenutrition is a feasible way to manage nutrition, more extensive studies are needed to ensure practical utility and effectiveness in utilizing various software and applications, such as telenutrition, to manage nutrition in this population.

### 4.1. Software Applications Dominance

The prominent area of technology found in the extant research reviewed was software. Sixty-eight percent (*n* = 31) of the studies used some form of software. The use of software falls within four areas that we label as (1) track, plan, and execute; (2) assessment; (3) build knowledge; and (4) social media. First, 45.16% of the studies use track, plan, and execute as the use of software to track dietary intake, plan for any changes, and execute actions in the plan. The software used was a mix of existing and existing software with modifications and custom development. All the software was for niche use, including a limited population and functional features. None of the articles included in this scoping review utilized commercially available software popular with everyday consumers. In the U.S., these software applications include Lose It! [90], Fooducate [91], and MyPlate [92]. In many cases, these apps are free; however, there is usually a cost for tracking important dietary intake goals for older adults, such as fluid, protein, and sodium intake. Of concern, the typical app promotion is for weight loss [93,94], which could create some confusion for older adults since, in some older adults, weight gain is most important. Nevertheless, using software to track, plan, and execute a nutrition program is essential.

The second area of software used in the studies is assessment, at 35.48% of the studies. Assessment involves obtaining data through a provider’s use of technology to support an older adult or an older adult’s self-use of technology. In this study’s papers, many researchers converted existing forms to apps or web pages, which may be a future mandate for nutritionists as data collection requirements expand. Government legislation and regulation required the wholesale collection and sharing of health care data, as seen in the U.S. HITECH Act, 2009 [95]. In that act, electronic medical records (EMRs) are a clinical understanding of health care information between health care systems, networks, and providers and are widely used. Electronic health records (EHRs) include medical history, medications, treatment plans, etc., and are not as common in day-to-day use today. Nutrition assessment, monitoring, and evaluation are essential components of the nutrition care process to detect and treat poor nutritional status among older adults and understand intervention efficacy [8].

Our findings reveal that technology may be a promising way to expand healthcare providers’ ability to identify older adults with poor nutritional status or at nutritional risk [61,77].

More research is needed to understand how utilizing these applications to track nutritional status and diet can help reduce the number of adults at nutritional risk with undetected malnutrition and how technology can help improve nutritional status.

However, collecting and addressing the requirements for older adult data is no small task with or without EHRs. That said, sixty percent of older adults have two or more chronic medical conditions [96]. They often receive transition care for their needs [8], which creates a perpetual necessity to update data for many healthcare needs. Today’s remedies for collecting these data include an ongoing requirement for older adults to answer questions for those filling out or completing manual and electronic forms themselves. The Academy of Nutrition and Dietetics recommends that nutrition practitioners improve evidence-based outcomes to advance the relevance of nutrition programs [18]. Likewise, the NCP provides a systematic framework for tailored, high-quality nutrition care that considers clients’ values, needs, and evidenced-based recommendations [97]. The ongoing need rests in understanding that older adults have individualized diets based on nutrition needs, weight, chronic illness, medical complexities, and involvement in their diet plans [8]. Against this background, apps for evidence-based assessment will expand and need to be standardized.

Third, 16.13% of the studies use software to build knowledge, which is software to develop a provider or older adult’s understanding of nutrition. The intent is to increase an older adult’s learning on how nutrition can positively affect QOL. However, a health improvement may not be seen; underlying hindrances may be due to nutrition-related health issues, such as appetite loss and eating difficulties [61]. Yet, new visual techniques may help build knowledge for older adults and providers. Three-dimensional animation software has shown early promise to replace printed material for older adults [52]. Access to snips of information, video, and text helps caregivers remedy dysphagia [74]. Many of these software applications recognized the heterogeneity in our aging population and the vital need for personalized dietary guidance and messaging to improve outcomes [54,59,60]. As a result, while standardization of app interfaces may be needed to enhance usability for older adult end users, the nutrition content should be tailored to meet their health needs. While software to build knowledge was not prominent, future research may show that knowledge-building is inherent in tracking, planning, and executing software.

Finally, at 3.23% of the studies using software, social media uses electronic networks to support a healthy life. Online social communities for older adults are a venue for discussing nutrition, exercise, and mental health. Research has indicated that in nutrition, online communities promote topic discussion on aging where issues with diet develop [75]. Applications for sharing views and information, such as in social media committed to older adult health, safety, and welfare, are of value to those more information technology-minded.

### 4.2. Tablet Computers Dominance

The second dominant area of technology used was a tablet computer; of the studies reviewed, that number was 52.27% (*n* = 23). The primary use of tablets was for older adults’ SUT. Following in structure with the areas of software we labeled earlier, a tablet was used to track, plan, and execute within nine studies. Additionally, using a tablet for assessment was included in nine studies and to build knowledge was a part of three studies. Based on software as the predominant technology, it is unsurprising that the tablet computer was number one in the hardware category. The final use for a tablet computer was webcam capabilities for taking pictures of food before and after dining and videoconferencing. Photographing a meal allows an older adult to record the nutrition and calories consumed in the track, plan, and execute nutrition program. A tablet webcam was used for videoconferencing by older adults to complete support calls regarding nutrition follow-up or assistance. In one specific study, providers used a tablet with specialty software to build knowledge supporting older adults. Video and 3D knowledge-building resources for providers and older adults are a ready interactive opportunity.

Given the user-friendliness of tablets as well as the extensive device features, this type of hardware holds promise to assist with various steps of the NCP, including connecting older adults with providers for nutrition assessment and counseling, actively tracking health information, and allowing older adults to set and strive to achieve tailored dietary goals. Previous research has also noted that tablet computers are the most intuitive and user-friendly for older adults [98].

### 4.3. Notable Mention Technologies

Our findings reveal that researchers used hardware including tablets, desktops, laptops, and mobile or smartphones in 88.64% of the studies. This finding is significant because the legacy technologies of desktop, laptop, mobile, or even smartphones are not the leading technologies in current research. Based on the discussion above, tablet computers are the most-used hardware when conducting recent research with older adults. Previous research has noted that tablet computers are the most intuitive and user-friendly for older adults [96]. User-friendly acceptance indicates that investigations should be open to understanding the effects of newer technology and its adoption by older adults in the future since tablets may not continue as the prevailing technology. As technology changes, large-screen smartphones may prevail so that older adults maintain one device for voice, video, and internet application engagement.

Regarding the Internet of Things (IoT) family, they appeared in 56.82% of the studies. “IoT consists of objects embedded with technology that can sense or capture information, communicate over the internet, and interact with its features or outside influences [51].” The breakdown of studies extracted is as follows: home-based sensors (13.64%), smart devices (13.64%), mobile device sensors (11.36%), and fitness devices (18.18%). There seems to be a movement toward conducting more research on the IoT, such as fitness and sensors; this is evident in current studies conducted from 2018 to 2020. The disciplines—IS, nutrition, and gerontology—are likely to conduct more research on the value of sensors in supporting older adults. Tracking and monitoring bring value by automatically assessing and reporting care and QOL measures [6,76].

We have seen the importance of the internet in day-to-day life across different age groups. While most of these studies did not explicitly mention internet use, internet use is pervasive. It will be integral in any investigation where two or more people are linked to complete a task or transact using digital services. The pandemic has highlighted the lack of broadband internet access to homes and small businesses nationwide. In particular, older adults living in rural areas lack digital services [99]. And proper internet access has become a necessity that U.S. legislators are interested in solving, as is evident in current federal law considerations [100]. Increasing broadband internet access for older adults can help offer digital nutrition services and help decrease social isolation in both community-dwelling older adults and those residing in long-term care communities experienced during the COVID-19 pandemic [101]. Research in videoconferencing will help bridge how telemedicine, telenutrition, and socialization enhance an older adult’s QOL.

## 5. Implications

Our scoping review reveals that nutrition and technology are needed to support an older adult’s QOL. The literature shows that software standardization is an essential initial step to being able to track, plan, and execute nutrition programs. Practitioners may not see existing software as transparent or simple enough for older adults to log and photograph food as a clear intake record. The value of standardization is making software use habitual as older adults maintain their nutrition across day-to-day and transition care for rehabilitation. Second, under the topic of software is the standardization of assessment software. When evident-based programs are needed, and the reporting values of a track, plan, and execute software are not detailed enough, standardized assessment software would be of value. The software input must be straightforward for the provider’s assessment evidence area and simplistic for older adult entry.

This current study’s findings rest within four areas: (1) software for track, plan, and execute and assessment functions lack standardization; (2) the family of the IoT is a promising area for new research for older adults; (3) personal device use by older adults appears to be evolving to the tablet computer; and (4) broadband internet is a technology source to health and nutrition care. Researchers in countries on continents worldwide, such as Australia, Asia, Europe, and North America, were involved in the findings revealed. As the world population ages, research is needed to build knowledge on older adults’ nutrition needs. Like innovation in any field, technology will be a part; however, how the end user responds to a technology’s capabilities is imperative to the innovation’s success.

Against this background, we find that only 51.1% of the older adult end users are engaged in the self-use of technology (see SUT percentage in Figure 3). The remaining use of technology, found in this scoping review, is the provider use of technology in support of the older adult, a combination of old adult self-users and provider users together, and providing for older adults’ ability or readiness to use technology. The expectations are that while the use of technology by providers of older adults is routine during their care, additional effort is necessary to place technology in the hands of the older adults. While these implications are significant in their own right, considering this research’s practical utility and effectiveness is also due for discussion.

### Implications for Practical Utility and Effectiveness

Society has become aware of the daily effectiveness of technology for almost all of life’s matters to enhance the average person’s QOL. Effectiveness with technology is ubiquitous in a world of all devices, always on, with multiple modality access to the internet. That said, technology integration is its users adopting the technology designed and deployed by hardware and software companies. However, older adults and those with disabilities adapt to technology since change is often required for older adults to use some hardware and software.

Regarding older adults, in an ad hoc analysis of the practical utility and effectiveness of nutrition and technology, we found in the papers for this study that most of the extant research was formative. We analyzed 12 studies where outcome data were reported from an intervention or randomized control trial to assess digital health’s practical utility and effectiveness for managing nutrition in our aging population. Only 11 (25%) of the retained studies were interventions [52,56,57,58,59,61,67,82,83,84,86], and one (2.27%) was a randomized control trial [88]. Eight studies examined used specially designed, proprietary software (customized software), rather than software readily available to the public, to assess or monitor nutrition status or provide nutrition education [52,58,59,61,67,80,82,85]. The other four studies used videoconferencing with older adults to provide education and counseling [56,57,86,88]. Overall, the primary outcomes of these 12 studies also varied, ranging from feasibility, acceptability, adherence, or changes in health markers pre- and post-intervention, which included weight status, nutrition knowledge, and lab values. While most studies support using technology to manage health, generalizing these findings is limited by the small sample sizes in the 11 intervention studies, ranging from 6 to 94 participants. Therefore, capturing the effectiveness of videoconferencing and customized software applications in improving older adults’ nutrition markers is difficult. Additionally, interventions are needed to assess adopted versus adaptive technology use, as well as to explore other hardware devices to manage health. Of these 12 studies extracted, only 4 examined fitness devices in conjunction with videoconferencing or software applications [56,57,59,80].

The practical utility of hardware and software is evident in the expanding integration into nutrition and healthcare management. As a result, more studies and support services research will help older adults benefit from these technologies and ensure practical utility in this population. Studies and services are time-sensitive since access to a hardware device connected to the internet is required to take advantage of certain services, including government-funded programs. For example, older adults were a priority to receive the COVID-19 vaccine first; however, registration was only mainly online when the vaccines were first rolled out to the public. Many older adults did not have the digital competency to schedule an appointment at a vaccine site [102], highlighting gaps in the digital divide between younger (adopters of technology) and older adults (adaptors of technology) and how this divide can result in health inequities [102]. Finally, some older adults have reported forced adoption to utilize the patient portal by their physicians [103]. It is imperative that further research is conducted to ensure the practical utility, adoption, and effectiveness of technology to manage nutrition and promote health among older adults and those who provide care for our aging population.

## 6. Future Research and Limitations

Further research is needed, because of COVID-19, to realize better the movement to videoconference technology for health care digital information and support. The studies reviewed were pre-pandemic, and videoconferencing was considered a worldwide business technology and became an emerging healthcare technology. With advancements in technology and its use in health care, videoconferencing is destined to be the omnipresent software in one’s browser. Similar to tablet hardware, this popular consumer-voted software will become the leader. It will be best for older adults to follow where younger family members lead on videoconferencing as with the tablet computer. Our search also reveals that few studies addressed the use of social media by older adults; however, the new generation of aging adults is technology-aware of the many personal devices, hardware, software, and social media available. This next generation of older adults will likely not submit to aging out of technology yet demand more age-specific technology. The use of social media is a promising area to introduce future older adults to videos, 3D experiences, and new learning to build knowledge.

How technology could assess dietary intake, improve diet quality, and food provision was also studied to a lesser extent. Only one study examined the efficacy of protein-enforced home-delivered meals selected from a tablet and monitoring intake for patients discharged from the hospital [67]. Scott et al.’s [78] findings reveal that smartphone applications can be used to access food delivery services. Given that the COVID-19 pandemic has disrupted how we traditionally purchase food in the grocery store [104], future studies should examine how to train older adults to use online grocery shopping to buy nutritious foods that meet their unique nutritional needs. Whether or not purchasing groceries online impacts diet quality among older adults should also be examined. Digital technology for food provisioning has increased the utilization of meal delivery kits (delivery of pre-portioned, fresh ingredients and recipes); how these kits among older adults improve diet quality with the help of an RDN should also be explored [105]. Additionally, since a critical contributor to inadequate dietary intake among older adults is decreased food enjoyment and food involvement (desire to prioritize food [9]), how these services improve these risk factors should be examined.

Future studies involving any new investigation should be considered open to various research methods. Further research should include a focus on increasing the number of participants. While data are more obtainable when using secondary data sources, participant group size can be an issue for new face-to-face studies. Older adults are susceptible to exclusion from studies for reasons not typically found in younger age groups, such as corrective eyesight limitations, increased chronic illness, and loneliness due to social isolation [106]. These limited population studies are critical since older adults have similarities yet are very heterogeneous in their medical and nutrition needs.

Findings from this research can be incorporated into the Theory of Andragogy [107] to help inform the design of technologies and telenutrition programs for older adults and healthcare providers. The Theory of Andragogy recognizes adult learners as mutual partners in learning, acknowledging that their prior experience can help them learn a new skill. When educating older adults and health professionals on using technology to manage nutrition, we suggest training sessions based on learning objectives that fulfill the end user’s requests, interests, and digital competence levels.

## 7. Conclusions

Our scoping review provides valuable evidence of the extant literature in the discipline of technology, nutrition, and geriatrics. The time frame of the literature search was the last 21 years; however, the search results show that 86% (*n* = 38) of the studies are within the five years from 2016 to 2020. This study time frame reveals the broader understanding that research interlocked in technology, nutrition, and geriatrics is new. And during this period, the critical analysis of the findings reveals that software for older adults to track, plan, execute, and assess evidence-based nutrition programs needs standardization. The IoT is a promising area for new research in QOL, and personal device use appears to evolve to the tablet computer. Finally, broadband internet is a vital nutrition care technology source. The results suggest that research on older adults’ nutrition using technology is not yet a formable research area; however, building knowledge is underway.

## Figures and Tables

**Figure 1 nutrients-15-04402-f001:**
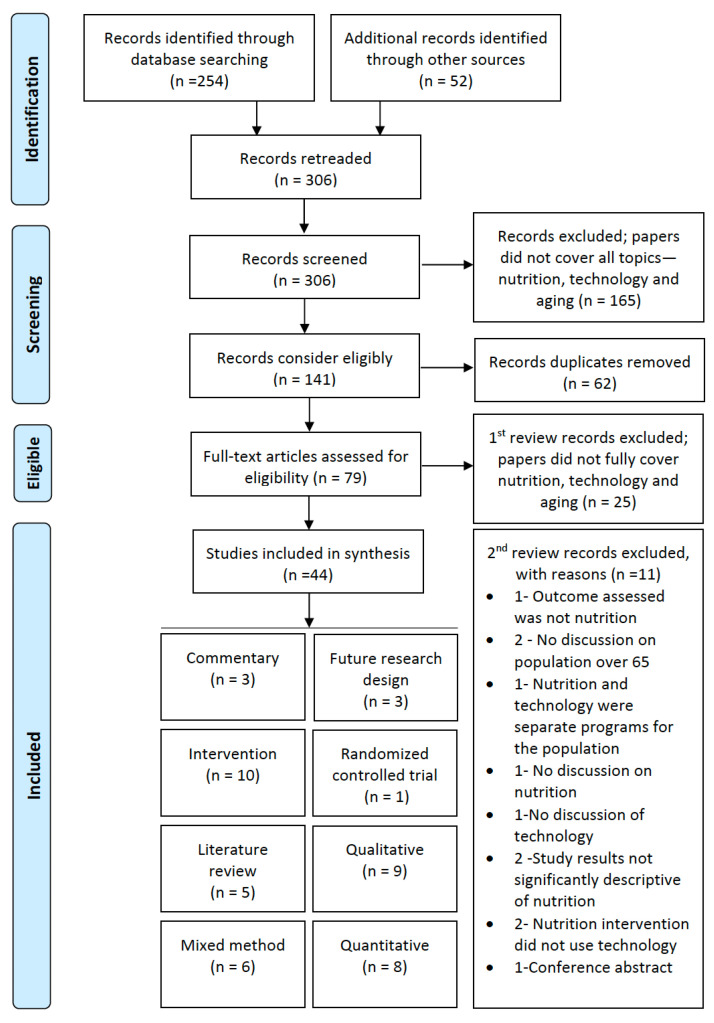
Process of identification and inclusion in this study: PRISMA diagram flow.

**Figure 2 nutrients-15-04402-f002:**
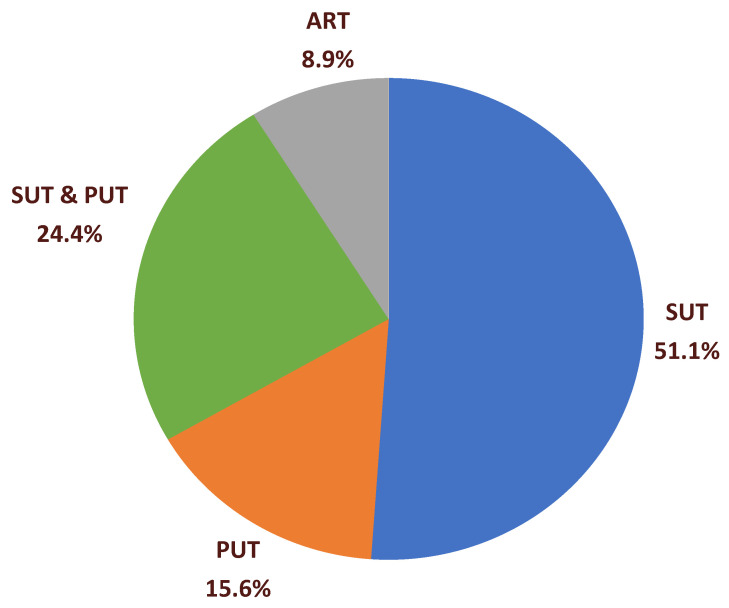
Target end users of technology.

**Figure 3 nutrients-15-04402-f003:**
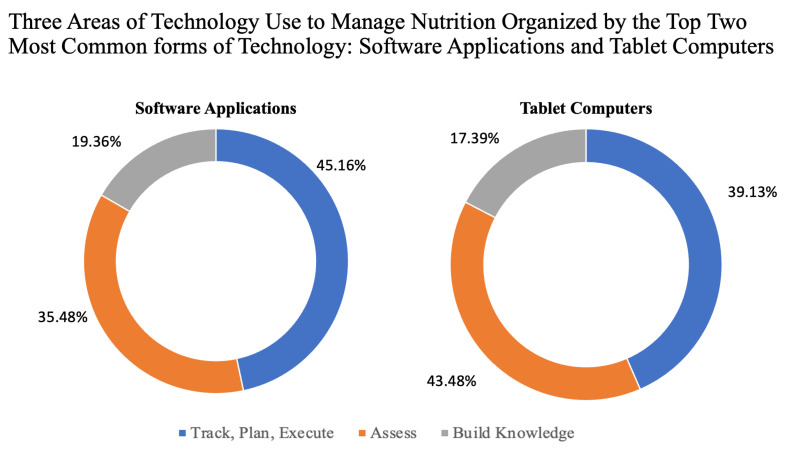
Prominent uses of software applications and tablet computers.

**Table 1 nutrients-15-04402-t001:** List of nutrition and information systems (IS) journals included in the initial search.

Nutrition Journals	IS Journals
American Journal of Clinical Nutrition	European Journal of Information Systems
Journal of Nutrition Education and Behavior	Information Systems Journal
Journal of the Academy of Nutrition and Dietetic	Information Systems Research
The Journal of Nutrition	Journal of Association for Information Systems
The Journal of Nutrition in Gerontology and Geriatrics	Journal of Information Technology
Nutrients	Journal of Management Information Systems
	Journal of Strategic Information Systems
	MIS Quarterly

## Data Availability

All data available upon request.

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
