# Peer review of "A Scoping Review of Nutrition Health for Older Adults: Does Technology Help?"

_nutrients, 2023, doi:10.3390/nu15204402_

Round 1
Reviewer 1 Report
Several issues should be revised before considering for acceptance of this review paper.
1. The “Abstract” section introduced the definitions of different technologies and their application areas, which may not be the important things of this paper. The novel findings after comparing the previous studies and analyzing the data should be described comprehensively. I suggested that this part should be rewritten
2. The presentation of the nutrition journals and the information systems journals should be in the tables or figures, not listed in the context.
3. When comparing the advantages and disadvantages of the previous studies using technology for managing nutrition for older adults (Table 1), the authors used a lot of sentences to describe the purpose and outcomes. It was too tedious, and the main points were not clear. Please use some short sentence or phrase to present these information.
4. The authors used a lot of abbreviations through the whole paper, however, some abbreviations (such as PRISMA-ScR, ABI/INFORM, and EMR) were not shown their full names. The abbreviation should be used the full name when it first appeared. Please check and revise this point.
5. The results section should be shortened and the discussion section should be expanded.
6. Please check and unify the format of references in accordance with the guidelines of “Nutrients”.
Reviewer 2 Report
Thank you for the opportunity to read this well-written, clear and concise study. It was interesting and a pleasure to read.
Title
- The type of study carried out must be entered.
Introduction
This section is very complete and allows the reader to know the main topic of the research, informs about the purpose and importance of the work in the clinical field, and also answers the question posed in the scientific context. It includes previous works on the topic in question and makes clear the aspects to be detailed in the review, which constitutes the object of the proposed research. It explains the general problem of the research, includes previous work on the topic in question, and specifies the objective of the study.
Methods
It is one of the most fundamental sections of a scientific article with these characteristics and is well developed and organized. The research method used to locate relevant studies is exhaustive and the design used for the selected articles is appropriate. However, some aspects should be reviewed, as they could alter the veracity of the study, as follows:
- In the first paragraph of this section, the PROPERO registration number of the study must appear.
- It would be convenient to add from and to the date on which the systematic search was carried out. This is important so that the reader knows the latest work carried out on the topic in question.
The results obtained are precise, so the review can be characterized by external validity and clinical importance. Although the main characteristics of the selected studies are presented in Table 1, it is recommended that information be added about the study variables and their measurement instrument.
The other sections of the review are considered adequate, except for the references section, which must adapt to the journal's standards. Please modify this section.
Reviewer 3 Report
1. This study conducted a comprehensive review of a substantial number of papers and provided an overview of the current state of technology in the context of nutrition for the elderly, along with potential future directions. However, it is important to note that the results should also include an assessment of the practical utility and effectiveness of employing these technologies.
-
2. The study did not specify the criteria used to identify the top journals in the fields of nutrition and information systems. It is essential to include information regarding the criteria employed for selecting these top journals.
-
3. In Figure 1, there is a discrepancy in the total number of individual studies, which is described as 44, while the correct number should be 45. Please rectify this error to accurately reflect the total number of studies reviewed.
Round 2
Reviewer 1 Report
It can be accepted.